

# Formaldehyde evolution in U.S. wildfire plumes during

# FIREX-AQ

Jin Liao[1,2], Glenn M. Wolfe[1], Reem A. Hannun[1,3], Jason M. St. Clair[1,3], Thomas F. Hanisco[1],

Jessica B. Gilman[4], Aaron Lamplugh[4,5], Vanessa Selimovic[6], Glenn S. Diskin[7], John B. Nowak[7],

Hannah S. Halliday[8], Joshua P. DiGangi[7], Samuel R. Hall[9], Kirk Ullmann[9], Christopher D.

Holmes[10], Charles H. Fite[10], Anxhelo Agastra[10], Thomas B. Ryerson[4,*], Jeff Peischl[4,5], Ilann

Bourgeois[4,5], Carsten Warneke[4], Matthew M. Coggon[4,5], Georgios I. Gkatzelis[4,5,**], Kanako

Sekimoto[11], Alan Fried[12], Dirk Richter[12], Petter Weibring[12], Eric C. Apel[9], Rebecca S. Hornbrook[9],

Steven S. Brown[4], Caroline C. Womack[4,5], Michael A. Robinson[4,5], Rebecca A. Washenfelder[4],

Patrick R. Veres[4], J. Andrew Neuman[4,5]

[1]Atmospheric Chemistry and Dynamics Laboratory, NASA Goddard Space Flight Center,

Greenbelt, MD, USA

[2]Universities Space Research Association, Columbia, MD, USA

[3]Joint Center for Earth Systems Technology, University of Maryland Baltimore County, Baltimore,

MD, USA

[4]NOAA Chemical Science Laboratory (CSL), Boulder, CO, USA

[5]Cooperative Institute for Research in Environmental Science (CIRES), University of Colorado,

Boulder, CO, USA

[6]Department of Chemistry, University of Montana, Missoula, MT, USA

[7]NASA Langley Research Center, Hampton, VA, USA

[8]Environmental Protection Agency, Durham, NC, USA





[9]Atmospheric Chemistry Observations & Modeling Laboratory, National Center for Atmospheric
Research, Boulder, CO, USA
[10]Earth, Ocean and Atmospheric Science, Florida State University, FL, USA
[11]Yokohama City University, Japan
[12]Institute of Arctic and Alpine Research (INSTAAR), University of Colorado, Colorado, USA.
[*]now at Scientific Aviation, Boulder, Colorado, USA.
[**]now at Forschungszentrum Jülich GmbH, Julich, Nordrhein-Westfalen, DE, Germany.
Correspondence email: jin.liao@nasa.gov



Abstract
Formaldehyde (HCHO) is one of the most abundant non-methane volatile organic compounds
(VOCs) emitted by fires. HCHO also undergoes chemical production and loss as a fire plume ages,
and it can be an important oxidant precursor. In this study, we disentangle the processes controlling
HCHO by examining its evolution in wildfire plumes sampled by the NASA DC-8 during the
FIREX-AQ field campaign. In nine of the twelve analyzed plumes, dilution-normalized HCHO
increases with physical age (range 1 – 6 h). The balance of HCHO loss (mainly via photolysis)
and production (via OH-initiated VOC oxidation) controls the sign and magnitude of this trend.
Plume-average OH concentrations, calculated from VOC decays, range from $-0.5$ ($\pm$ 0.5) $\times 10^6$
to 5.3 ($\pm 0.7$) $\times 10^6$ cm$^{-3}$. Plume-to-plume variability in dilution-normalized secondary HCHO
production correlates with OH abundance rather than normalized OH reactivity, suggesting that
OH is the main driver of fire-to-fire variability in HCHO secondary production. Analysis suggests
an effective HCHO yield of 0.33 ($\pm$ 0.05) per VOC molecule oxidized for the 12 wildfire plumes.
This finding can help connect space-based HCHO observations to the oxidizing capacity of the
atmosphere.





1. Introduction
Wildfire biomass burning is a large source of trace gases and aerosols that affect regional
atmospheric chemistry, human health, air quality, radiative balance and climate. Wildfire
frequency and intensity are expected to increase with global warming under higher temperatures
and drier conditions in the future (Westerling et al., 2006). Wildfire emissions of volatile organic
compounds (VOCs) are a complex mixture spanning orders of magnitude in concentration,
reactivity, and volatility (Gilman et al., 2015; Koss et al., 2018). These VOCs contribute to
increased regional tropospheric ozone (Alvarado et al., 2010; Jaffe and Wigder, 2012; Mauzerall
et al., 1998; Wotawa and Trainer, 2000) and can deposit onto or evaporate from organic aerosols
in biomass burning air masses (Garofalo et al., 2019; Majdi et al., 2019; Palm et al., 2020).

Formaldehyde (HCHO) is one of the most abundant non-methane VOCs emitted by wildfires
(Akagi et al., 2011; Gilman et al., 2015; Simpson et al., 2011). HCHO emissions vary with total
carbon emissions, modified combustion efficiency (MCE) and fuel type. Emission factors of
HCHO decrease as MCE increases (e.g., Liu et al., 2017; Yokelson et al., 1999), indicating that
more HCHO is produced from smoldering fires than from flaming fires. HCHO emissions can
vary by more than a factor of 2 among tropical forest, savanna, boreal forest and temperate forest
biomes (Akagi et al., 2011). In addition to direct emissions, HCHO is formed in fire plumes via
VOC oxidation. Alvarado et al. (2020) used TROPOMI data to show that HCHO enhancements
in wildfire plumes persist for days downwind. HCHO also serves as an important source of peroxy
radicals ($HO_2$), thereby influencing the formation of ozone and other secondary pollutants
(Yokelson et al., 1999).




Few studies have investigated the photochemical evolution of HCHO in biomass burning plumes,
and these studies have reported both net HCHO production and loss. Mauzerall et al. (1998)
reported average HCHO enhancement (ΔHCHO/ΔCO) of 9.5 ppt/ppb for fresh plumes (less than
0.5 day), 1.8 ppt/ppb for recent plumes (less than 1 day), 2.3 ppt/ppb for aged plumes (< 5 days
old), and 0.9 ppt/ppb for old plumes (>5 days old). Trentmann et al. (2005) observed a potential
increasing trend of ΔHCHO/ΔCO from 20 ppt/ppb to over 30 ppt/ppb with limited data and
simulated a flat trend of ΔHCHO/ΔCO within 1 h age since emission from a Savanna fire plume
in Africa. Müller et al. (2016) also observed an increasing trend of ΔHCHO/ΔCO with an average
of 22.7 ppt/ppb and simulated a flat or slightly decreasing trend of ΔHCHO/ΔCO in a small fresh
agricultural biomass burning plume in Georgia, US. While such case studies are valuable, we lack
a general understanding of the drivers of plume trends and plume-to-plume variability in HCHO
evolution.

HCHO is also one of the few VOCs that can be observed from space, and the global coverage of
satellite observations has been leveraged to provide insights into a variety of atmospheric
chemistry questions. HCHO is correlated with organic aerosols in biomass burning air masses, and
this correlation might be exploited to estimate organic aerosol abundance from satellite HCHO
measurements (Liao et al., 2019). In regions with constant or very high OH reactivity, HCHO
variability is closely linked to OH variability (Valin et al., 2016; Wolfe et al., 2019) and may be
used to infer OH. Satellite HCHO columns have also been widely used to constrain emissions of
isoprene and other VOCs (Fu et al., 2007; Kaiser et al., 2018; Marais et al., 2014; Millet et al.,
2008; Stavrakou et al., 2009). Understanding the emissions, chemistry and trends of HCHO in



wildfires will facilitate the application of satellite HCHO towards broad-scale wildfire smoke
processes and impacts.

The Fire Influence on Regional to Global Environments and Air Quality experiment (FIREX-AQ)
deployed a comprehensive suite of instruments aboard the NASA DC-8 aircraft to study wildfires
and agricultural fires in the US. It provided a great opportunity to systematically study the
emissions and chemistry of HCHO in wildfire plumes. In the following, we describe the HCHO
dependence on plume age in wildfire plumes from FIREX-AQ, assess the drivers of HCHO trends,
and examine the factors controlling variability in secondary HCHO production.

2. Methods
2.1 FIREX-AQ field campaign and measurements description
During FIREX-AQ, a combination of four aircraft (the NASA DC-8, NASA ER-2, and two
NOAA Twin Otters) with a comprehensive suite of in situ and remote sensing instruments were
deployed to characterize fire emissions and chemistry with operational bases in Boise, ID and
Salina, KS from July to September 2019. This study focuses on wildfire plumes sampled by the
NASA DC-8 aircraft during FIREX-AQ.

In situ HCHO observations were acquired by several instruments onboard the DC-8; here we
primarily use measurements from the In Situ Airborne Formaldehyde (ISAF) instrument (Cazorla
et al., 2015). ISAF uses laser-induced fluorescence to detect HCHO. A tunable UV laser excites
HCHO molecules to an excited electronic state and the resulting fluorescence is detected with a
photon-counting photomultiplier tube. The laser wavelength is modulated on and off a rotational





absorption feature (353.163 nm), and the difference between the "online" and "offline" signals is
proportional to the HCHO concentration.

ISAF was calibrated pre- and post-mission with a compressed-gas HCHO cylinder (584 ± 15 ppbv
in nitrogen, Air Liquide). Sensitivity typically varies by less than 5% between calibrations. Flow
meters for the standard dilution system were calibrated against a DryCal calibrator (Mesa Labs)
with an accuracy of < 1%. The HCHO standard concentration was calibrated before and after the
field deployment with an MKS Multigas 2031 Fourier transform infrared spectrometer. Gas
standard mixing ratios are typically reproducible to within 2% of the mean value measured over
multiple years. IR-determined mixing ratios are adjusted by a factor of 0.96 based on a separate
long-path UV absorption experiment (Cazorla et al., 2015). Thus, ISAF HCHO mixing ratios are
ultimately tied to the UV cross sections of Meller and Moortgat (2000) as recommended by the
JPL 2011 evaluation (Sander et al., 2011). The detection limit of ISAF was 30 pptv for 1-Hz data
at signal/noise = 1 and the accuracy of ISAF HCHO measurements was estimated as 10% + 10
pptv. The 1/$e$ response time of ISAF during FIREX-AQ was about 300 ms, limited mainly by
flow through the sample cell.

During FIREX-AQ, ISAF HCHO measurements correlated with those from the Compact
Atmospheric Multispecies Spectrometer (CAMS) (Richter et al., 2015), with a correlation
coefficient $r^2$ of 0.99, a slope of 1.27 (CAMS vs. ISAF), and a near-zero intercept for 1-Hz average
wildfire data from equally weighted orthogonal distance regression (Fig. S1). The systematic bias
between the CAMS and ISAF measurements exceeds the combined stated uncertainty (10% for
ISAF, 6% for CAMS). Post-mission comparisons suggest this discrepancy is due to the absolute





calibration of compressed-gas HCHO standards, which are tied to literature-recommended UV
(ISAF) or IR (CAMS) cross sections; the source of this discrepancy is still under investigation.
Remotely-sensed HCHO column retrievals rely on the same UV cross sections (De Smedt et al.,
2018) that are used to calibrate the ISAF instrument. The HCHO enhancements in the plumes (Sect
3.1) and the estimated effective yield of HCHO from VOC oxidation by OH (Sect. 3.3) can have
a potential low bias of 27% due to the ISAF and CAMS HCHO measurement difference. This
uncertainty proportionally affects quantitative analysis results but does not alter qualitative
conclusions.

We also use several supporting measurements in our analysis. CO was measured via mid-IR
wavelength modulation spectroscopy by the Differential Absorption Carbon Monoxide
Measurement (DACOM) instrument (Sachse et al., 1991). Photolysis rates were derived from the
Actinic flux measurements by the Charged-coupled device Actinic Flux Spectroradiometer (CAFS)
(Hall et al., 2018). Alkenes were measured by the NOAA Whole Air Sampler (iWAS) (Lerner et
al., 2017). Ozone ($O_3$) measurements were from the NOAA Chemiluminescence instrument
(Bourgeois et al., 2020). OH reactivity calculations used VOCs measurements from the NOAA
Proton-Transfer Reaction Time-of-Flight Mass Spectrometry (PTR-ToF-MS) (Yuan et al., 2016),
NCAR Trace Organic Gas Analyzer (TOGA) (Apel et al., 2015) outfitted with a Time-of-Flight
Mass Spectrometer, NOAA Airborne Cavity Enhanced Spectrometer (ACES) (Min et al., 2016),
and NOAA Iodide Ion Time-of-Flight (ToF) Chemical Ionization Mass Spectrometer (CIMS)
(Veres et al., 2020), listed in Table S1.  Our analysis uses in situ measurements that are merged to
the iWAS sampling period, which ranged from 1-9 seconds per canister, such that multiple samples
were often acquired within a single plume crossing.




2.2 Normalized excess mixing ratio (NEMR) and physical age definitions
NEMR is defined as the difference between the concentration of species X in the plume and in the
background air outside of the plume, normalized by the difference between CO concentrations in
the plume and the background outside of the plume.
$NEMR = \frac{\Delta X}{\Delta CO}$ (1)
The background air outside of the plumes was manually selected and could be different or the
same for different transects of the same plume, depending on the availability of the iWAS data.
The HCHO NEMR is denoted by nHCHO below.

Physical age was estimated using a Lagrangian trajectory analysis (Holmes et al., in preparation)
and described briefly here. Fire source locations were pinpointed using the MODIS/ASTER
Airborne Simulator (MASTER) instrument data onboard the DC-8. Upwind trajectories from
aircraft locations were computed and the advection age was calculated from the time when a
trajectory was closest to the fire. Plume rise time from the surface to the trajectory initialization
altitude assumed a vertical wind speed of 7 m/s. The smoke age is the sum of advection age plus
rise age averaged over several meteorological models. The average uncertainty of the estimated
physical age for the analyzed wildfire plumes was 37% with an interquartile range of 20% based
on the range of ages derived from the High-Resolution Rapid Refresh (HRRR), North American
Mesoscale Forecast System (NAM) CONUS Nest, and Global Forecast System (GFS 0.25°)
meteorological datasets.

2.3 Plume selection





Details about the specific selected wildfire plumes among all sampled wildfire plumes during
FIREX-AQ are provided in Table S2. Wildfire plumes that meet the conditions listed below are
selected to study the evolution of HCHO in wildfires.
a) Lagrangian sampling patterns
Lagrangian sampling patterns are defined as flight tracks intercepted the plumes with flight leg
directions approximately perpendicular to the horizontal wind directions and more than three
transects downwind with different distances from the fire.
b) Enhanced HCHO mixing ratios above background
We selected the plumes with maximum 1-Hz HCHO mixing ratios > 600 ppt, which was close to
the ambient background HCHO mixing ratios. The North Hill plume on 29 July 2019 is the plume
with the lowest HCHO concentrations among the selected plumes.
c) Appropriate VOC decay for the period analyzed with sufficient data samples
We selected the plume samples where chemical age correlated with physical age. This was defined
by a correlation coefficient $r^2 \geq 0.57$ for a plot of ln(trans-2-butene/propene) or ln(cis-2-
butene/propene) vs physical age. We used 2-butenes/propene as chemical age tracers in this
analysis because these gases have comparable lifetimes to physical age for most of the analyzed
plumes. We filtered out plume data if the correlation coefficient of ln(trans-2-butene/propene) or
ln(cis-2-butene/propene) vs. physical age degraded at older physical ages. Figure S2 shows
ln(trans-2-butene/propene) and ln(cis-2-butene/propene) vs. physical age for the plumes that
satisfied conditions a) and b) and had iWAS data available. The threshold of $r^2 = 0.57$ is chosen
by visual inspection of all VOC decay in Fig. S2.  We also filtered out plumes with total number
of data points < 8 in the iWAS sample periods for an entire selected circuit of multiple plume
transects with good VOC decay. Due to the inhomogeneity of the plumes, too few data points can





introduce large bias.  In the analyzed plume periods, ln(trans-2-butene/propene) or ln(cis-2-
butene/propene) also has good correlations with the maleic anhydride/furan ratio (Fig. S3), another
tracer of chemical age in biomass burning plumes (personal communication with Carsten Warneke
and Matthew M. Coggon, 2021). The Mica and Lick Creek plume on 02 August 2019 is the plume
with the least number of data points among the selected plumes (N = 8).

The above filters, applied to a total of 26 fire plumes, yield 11 daytime plumes and 1 nighttime
plume that are suitable for our analysis (Table S2). One of the twelve plumes (Blackwater)
occurred in the southeast US and the rest eleven plumes were in the western US.

2.4 Estimating average OH concentrations in the plumes
Plume photochemical age is estimated based on the relative decay of primary emitted VOCs that
have different reaction rate coefficients with OH (e.g., Warneke et al., 2007). We can estimate the
average concentration of OH by combining the photochemical age with the trajectory-based air
mass age. Cis-2-butene/propene ratios and trans-2-butene/propene ratios are used to estimate OH
in this analysis because these gases have comparable lifetimes to physical age (2–6 h) for most of
the analyzed plumes. The lifetimes of propene, cis-2-butene, and trans-2-butene are approximately
4.5 h, 2.3 h, and 1.8 h, respectively, at OH concentrations of $2 \times 10^6$ molecules cm$^{-3}$ (Atkinson et
al., 2006). Because both 2-butenes also differ from propene in $O_3$ reaction rate coefficients, the
reactions of these alkenes with $O_3$ are also considered when we estimate the OH concentrations.
We assume that the variability in the butenes–propene relationship is driven by OH and $O_3$ and
that there is negligible change in the relative emission ratios over the sampled plumes. These
reaction rate coefficients are those reported by Atkinson et al. (2006).



$\ln \frac{\text{butene}}{\text{propene}} = \ln \frac{\text{butene}_0}{\text{propene}_0} - \{(k_{\text{butene}+\text{OH}} - k_{\text{propene}+\text{OH}})[\text{OH}] + (k_{\text{butene}+O_3} - k_{\text{propene}+O_3})[O_3]\}t$

232 (2)

OH concentrations are derived from the slope of $\ln \frac{\text{butene}}{\text{propene}}$ vs. $t$ (physical age), the measured ozone
concentrations and the reaction rate coefficients.
$[\text{OH}] = \frac{slope_{butene} + (k_{\text{butene}+O_3} - k_{\text{propene}+O_3})[O_3]}{k_{propene+OH} - k_{butene+OH}}$ (3)
The average ozone concentration of the entire circuit with multiple transects is used to represent
the integrated $O_3$ effect on alkene oxidation. The uncertainty due to $O_3$ variation and the
uncertainty in the slope of $\ln \frac{\text{butene}}{\text{propene}}$ vs. $t$ are propagated to estimate the total uncertainty in plume-
average OH.

2.5 Calculating primary HCHO normalized mixing ratios and secondary HCHO production rates
To understand the relative importance of primary emission vs. secondary production of HCHO in
fire plumes downwind, we calculate primary and secondary HCHO as the plume ages. The primary
HCHO time profile is calculated by the following equation:
$\text{nHCHO}_{\text{primary}} = \text{nHCHO}_0 \exp(-(J_{HCHO} + k_{HCHO}[\text{OH}])t)$ (4)
where $\text{nHCHO}_0$ is equal to the fitted observed nHCHO (HCHO NEMR) closest to the fire source,
$J_{\text{HCHO}}$ is the measured HCHO photolysis frequency in iWAS sample periods averaged and
interpreted in physical age space, $k_{HCHO}$ is the reaction rate coefficient between HCHO and OH,
and $t$ is the physical age. $\text{nHCHO}_{\text{secondary}}$ is calculated by subtracting $\text{nHCHO}_{\text{primary}}$ from the
measured nHCHO. Here we assumed the fitted observed nHCHO closest to the fire source is equal
to nHCHO at the emission source. This assumption will not impact the secondary nHCHO
production rate calculated below.

To characterize secondary HCHO production in wildfire plumes, we calculate the secondary
nHCHO production rate. The secondary nHCHO production rate is derived from the HCHO mass
balance equation.
$\frac{d\,HCHO}{dt} = P - L - D$ $\qquad\qquad$ (5)
where P is chemical production, L is chemical loss, and D is dilution. The calculation of the
secondary nHCHO production rate is shown in eqn. 6. The derivation of eqn. 6 from eqn. 5 can be
found in the Appendix A.
$\frac{P}{\Delta CO}$ (or $P_{nHCHO}$) $= \frac{dnHCHO}{dt} + (J_{HCHO} + k_{HCHO}[OH])nHCHO.$ $\qquad$ (6)
Here, $\frac{d\,nHCHO}{dt}$ is taken as the slope of measured nHCHO vs physical age and other parameters are
as defined above.

2.6 Impact of potential variation in HCHO emission ratios on nHCHO trend
In this analysis, we assume the variability in the HCHO/CO emission ratio (that is, nHCHO at the
source) is much smaller than the variability in nHCHO induced by chemistry for any single fire
plume. Emission factors of both HCHO and CO (that is, g of gas per kg of fuel burned) depend on
MCE, fuel type, and other factors (e.g., Liu et al., 2017; Yokelson et al., 1999). Normalizing
HCHO by CO removes the strong negative dependence of HCHO emission factors on MCE.  A
small positive trend of nHCHO vs. MCE is due to higher nHCHO and MCE for the eastern US
wildfire plume than the western US wildfire plumes (Fig. S4). No clear trend of MCE in nHCHO
plume evolution was observed in FIREX-AQ data (Fig. S5). Emissions of $CO_2$ correlate with fire
radiative power (FRP) detected by satellite during FIREX-AQ, and the variability of FRP could
affect the variability of downwind concentrations (Wiggins et al., 2020). We found that HCHO



correlates with $CO_2$ (Fig. S6a) and thus likely also with FRP because the change of measured $CO_2$
correlates with the change of FRP (Wiggins et al., 2020). To account for emission variation and
dilution, which are main factors affecting the absolution concentrations of trace gases and aerosols
in the plumes, HCHO is normalized to CO to investigate the impact of photochemistry on HCHO
evolution in the plumes. Photochemistry takes place while emission varies. When normalized to
CO, nHCHO does not strongly depend on $CO_2$ (Fig. S6b and Fig. S7) or FRP. FRP and MCE do
not control the trends of nHCHO.

2.7 OH reactivity calculation
We calculate the observed OH reactivity using the Framework for 0-D Atmospheric Modeling
(F0AM v4) (Wolfe et al., 2016) with the Master Chemical Mechanism v3.3.1 (MCM; Jenkin et al.,
2015) and additional chemical reactions from recent publications of newly-observed biomass
burning species and reactions (Coggon et al., 2019; Decker et al., 2019). The VOC chemical
species included in the F0AM model are listed in Table S1. We calculate the OH-VOC reactivity
($\sum k_i VOC_i$) by excluding OH reactions with $NO_2$ and CO.

3. Results and discussion
3.1 Trends of HCHO in wildfire plumes
nHCHO in wildfire plumes can increase or decrease as plumes age. The trends of measured
nHCHO vs. physical age and the corresponding quadratic polynomial regression for 12 selected
plumes are plotted in Fig. 1. Quadratic polynomial regression is used because it has suitable
degrees of freedom to capture the trends. Considering the CO measurement uncertainty of $\leq 7\%$
and HCHO measurement uncertainty of 10%, the uncertainty of nHCHO is estimated to be $\pm 12\%$



with a potential systematic low bias of as much as 27% (based on the difference between ISAF
and CAMS). Random error due to HCHO and CO measurement precision is negligible when
averaging over the iWAS integration time in high-concentration biomass burning plumes.

In the absence of secondary production, we expect nHCHO to decay with a time constant of a few
hours in the daytime. The blue curves in Fig. 1 show the predicted decay of initial nHCHO using
observed HCHO photolysis rates and measurement-derived OH concentrations. Because the
variability in nHCHO in one transect is significant, we use the start point of the observed nHCHO
fitted curve to represent the observed nHCHO closest to fire. HCHO photolysis frequencies are
averaged over each transect and linearly interpolated to determine continuous age-dependent
photolysis frequencies. The calculated nHCHO without production represents an upper limit of
primary (emitted) nHCHO because some HCHO production and loss had already occurred before
the closest transect.

We can also estimate the primary nHCHO (black dashed curves) and the fraction of primary
HCHO by assuming nHCHO and the loss rate of nHCHO were constant between emission and the
closest observation.  The fraction of primary nHCHO to total nHCHO varies from plume to plume
and depends on secondary HCHO production rates and total HCHO loss rates. The primary HCHO
fraction could decay rapidly to be 60% in about 1 h of aging or it could decay slowly to still account
for 60% in about 5h of aging.  The primary and secondary fractions of HCHO indicate the impact
of direct emission and photochemistry on the fire plume composition downwind. The average and
standard deviation of nHCHO production and loss rates for each plume are provided in Table S3.





HCHO production exceeds loss in 9 of the 12 selected plumes, indicated by positive trends of
nHCHO vs. physical age in Fig. 1. Plumes exhibiting negative nHCHO trends have higher nHCHO
loss rates than production rates (Table S3). This shows that fire-to-fire variability in the overall
nHCHO trend relates to the balance between loss (via photolysis) and production (via VOC
oxidation). HCHO loss by photolysis can be either higher or lower than the loss by reaction with
OH, but on average photolysis is faster. HCHO loss via photolysis accounts for $63 \pm 27\%$ of the
total HCHO loss in daytime plumes. The average HCHO lifetime by photolysis was 8.2 ($\pm$ 8.8) h
for the 11 daytime plumes, shorter than the average HCHO lifetime by OH oxidation of 23.5 ($\pm$
31.3) h. For some plume transects, there was significant variability in HCHO photolysis
frequencies over iWAS averaging intervals due to the aerosol radiative effects. Applying filters to
only analyze the data with relatively homogeneous in-plume HCHO photolysis rates does not alter
our conclusions. Plume-average OH is not well correlated with the HCHO photolysis frequency
(Fig. S8), likely due to inter-fire variability of OH sources and sinks.

3.2 OH concentration estimation
OH is the main oxidant that reacts with VOCs to produce HCHO in the daytime. As described in
Sect. 2.4, we estimate plume-average OH concentrations using the relative decays of 2-butenes to
propene via eqn. 3. The decay of the natural logarithm of the trans-2-butene to propene ratio and
the cis-2-butene to propene ratio with physical age are plotted in Fig. 2 with significant correlation
($r^2 = 0.57$–0.99) for the 12 plumes. The lowest correlation coefficient occurs for the nighttime
plume on 12 August 2019 and the daytime plume on 29 July 2019. This indicates that the
photochemical age of the plumes is consistent with their physical age, and the oxidation chemistry
can be reasonably represented by average OH and $O_3$.






The estimated average OH concentrations for the 12 plumes are shown in Fig. 3. The uncertainties
in OH concentrations are based on the standard error in the slope of ln(butenes/propene) vs
physical age and the standard deviation of $O_3$ concentrations. The average and standard deviation
of $O_3$ concentrations and the uncertainty in OH estimation due to the impact of $O_3$ standard
deviation are listed in Table S4. The variation of OH concentrations derived from trans-2-butene
to propene ratios is generally consistent with that derived from cis-2-butene to propene ratios,
though OH concentrations from trans-2-butene have slightly higher (27% on average) values than
that from cis-2-butene to propene, which may be due to systematic bias in the reaction rate
coefficients at low temperature (276.9$\pm$3.9 K). The average OH concentrations from trans-2-
butene to propene and cis-2-butene to propene were used to represent the average OH
concentrations of the plumes. OH concentrations covered a large range, varying from $-0.5(\pm$
$0.5)\times 10^6$ (for a nighttime plume) to $5.3(\pm0.7) \times10^6$ molecules cm$^{-3}$.

3.3 Controls on secondary HCHO formation
The average secondary nHCHO production rate correlates with the average OH concentration ($r^2$
$= 0.69$, Fig. 4a). The secondary production rates of nHCHO were calculated from the trends of
observed nHCHO ($\frac{dnHCHO}{dt}$), photolysis loss rate and OH (eqn. 6). The uncertainty in nHCHO
secondary production rates for each plume is estimated from the standard deviation of the
calculated nHCHO secondary production rates along the physical age of the plume. The
uncertainty in estimated OH is determined by the propagated uncertainties of OH from trans-2-
butene to propene ratios and cis-2-butene to propene ratios. The good correlations ($r^2= 0.69$)
between the secondary production rate of nHCHO and average OH indicate that the variability in



OH is a key driver of the secondary production rate of nHCHO. Figure 4a is color-coded with
normalized OH-VOC reactivity calculated from measured VOCs (Sect. 2.7). Plume-average
normalized OH-VOC reactivity ranges from 11 to 31 $s^{-1}$ (ppm CO)$^{-1}$, is about 20% lower than total
OH reactivity across the analyzed plumes, and does not exhibit a clear relationship with OH. This
demonstrates that variability in OH, as well as secondary nHCHO production, likely depends
principally on variability in OH sources (e.g., photolysis of HONO and conversion of $HO_2$ by NO)
(Peng et al., 2020) rather than sinks. Because nHCHO trend, OH concentration, and normalized
OH-VOC reactivity all depend on physical age, in addition to the different properties of the plumes,
the difference in physical age among these plumes also has an impact on the average values.

Figure 4b shows nHCHO production vs. the product of OH and dilution-normalized observed OH-
VOC reactivity (averaged for each plume). The latter is a lower limit for the total average OH
loss/production rate as observations do not include all OH sinks. The correlation is slightly higher
than that in Fig. 4a because variability in normalized OH-VOC reactivity plays a smaller role than
OH in affecting $P_{nHCHO}$. The slope of this relationship, $0.33 \pm 0.05$, is a metric for the effective
yield of HCHO from OH-initiated VOC oxidation. Assuming that reaction of OH with a VOC is
the rate-limiting step and ignoring non-OH sources, integrated HCHO production can be written
as in eqn. 7.
$$P_{HCHO} = \sum \alpha_i k_i [OH] [X_i] = \alpha_{eff} k'_{OH} [OH] \qquad (7)$$
Where $\alpha_i$ is the yield of HCHO from OH oxidation of any VOC reactant $X_i$ and depends on both
the structure of X and the fate of reactive intermediates like peroxy radicals, $k_i$ is the reaction rate
coefficient for $VOC_i + OH$, $k'_{OH}$ represents VOC-OH reactivity, and $\alpha_{eff}$ is the effective yield
weighted over OH-VOC reactions. If all OH reactivity (including reactions with CO and $NO_2$)





instead of OH-VOC reactivity is considered, $\alpha_{eff}$ will be about 20% smaller. As discussed by Valin
et al. (2016), $\alpha_{eff}$ from all OH reactivity is expected to range from 0.2 to 0.4 depending on the
magnitude of $NO_x$ and the magnitude and speciation of VOC. The yield reported here (0.28 for all
OH reactivity) is on the low end of this range, implying that HCHO production in the plume is not
very efficient due to the nature of the emitted VOC and/or the balance of $RO_2$ reactions with NO,
$HO_2$, and other $RO_2$. High $\alpha_{eff}$ values reported by Valin et al. (2016) occur in high isoprene
emission regions, implying the emitted VOCs in wildfires are not as efficient as isoprene in
producing HCHO. Our $\alpha_{eff}$ of 0.28, when considered all OH reactivity, is higher than the value of
0.20 (± 0.01) derived by Wolfe et al. (2019) for total-column HCHO in the remote troposphere,
where methane oxidation is the primary HCHO source. The potential low bias in observed HCHO
could lead to a proportional (27%) low bias in in $\alpha_{eff}$. We used PTRMS $CH_3CHO$ measurements,
which had more complete data coverage than TOGA, and can be more easily integrated over the
iWAS sampling time than the TOGA $CH_3CHO$, which had a sampling duration between 12-33s
during FIREX-AQ, and was not always aligned with the iWAS time step. This could also lead to
slightly high bias (~5%) in the calculated OH reactivity and a low bias in the $\alpha_{eff}$. This indicates
that besides the potential missing VOCs, the uncertainties in measured VOCs concentrations also
contribute to the uncertainties in OH reactivity and $\alpha_{eff}$. The $\alpha_{eff}$ for the eastern US wildfire plume
seems to be higher than western US wildfire plumes but the uncertainties are large.  Higher
$NO_x$/VOC ratio in the eastern than western US wildfire plumes may contribute to the higher $\alpha_{eff}$
because more $NO_x$ generally means more radical turnover and a larger fraction of $RO_2 + NO$, both
of which favor HCHO production.

3.4 Implications for interpretation of satellite observations



The quantification of the evolution of HCHO in wildfire plumes can be leveraged to enhance
interpretations of satellite remote sensing observations. The good correlation of dilution-corrected
secondary HCHO production and oxidant levels suggests the use of satellite HCHO data to
estimate oxidant levels in biomass burning plumes. Similar to the studies of $NO_2$ lifetime from
satellite $NO_2$ data (e.g., Laughner and Cohen, 2019; Liu et al., 2016), with parameterized
production rates of HCHO as a function of OH from this study, the effective lifetime of HCHO
and OH concentrations in the wildfire plumes could potentially be derived from remote sensing
HCHO and CO data if the photolysis rates can be properly parameterized. Satellite HCHO
retrievals in biomass burning plumes remain challenging, and information about vertical
distributions of trace gases and aerosols from airborne measurements are likely needed to improve
satellite retrievals in biomass burning plumes. The effective yield of HCHO from this analysis
indicates that the biomass burning VOCs could be less efficient than isoprene in producing HCHO,
although other factors such as balance of $RO_2$ reactions with NO, $HO_2$, and other $RO_2$ can play a
role. This information may be useful for estimating VOC emissions from satellite HCHO data.


4. Conclusions
We studied the chemical evolution of HCHO in wildfire plumes during FIREX-AQ. Twelve well-
developed plumes with consistent chemical and physical age 1–6 h downwind were selected
among 26 wildfire plumes sampled. During plume transport and aging, dilution-corrected HCHO
increased in smoke from nine wildfires and decreased in three, depending on the balance of HCHO
production and loss processes. Secondary nHCHO production tracks average OH concentrations,
indicating that the variability in OH rather than the variability in the reactive VOC pool drives the





production of nHCHO in these wildfire plumes. The effective HCHO yield from OH-initiated
VOC oxidation is estimated to be 0.33 (± 0.05), which is about in the middle of previous studies
of isoprene-rich, urban VOC-dominated and remote atmospheric background regions.








Appendix A. Derivation of secondary nHCHO production rate from mass balance equation
Change of HCHO concentration with time can be obtained from mass balance equation (eqn. A1)
$\frac{d\,HCHO}{dt} = P - L - D$                 (A1)
where P is the HCHO chemical production term; L is the HCHO chemical loss term; and D is the
dilution term.
Considering the HCHO normalized excess mixing ratio ( $nHCHO = \frac{HCHO - HCHO_{bkg}}{CO - CO_{bkg}}$ ) and
assuming that the HCHO background change is relatively small ($\frac{d\,HCHO\,bkg}{dt} \approx 0$), $\frac{d\,HCHO}{dt}$ can be
written as
$\frac{d\,HCHO}{dt} = \Delta\,CO\,\frac{d\,nHCHO}{dt} + nHCHO\,\frac{d\,\Delta CO}{dt}$.              (A2)
Because L, D and P terms are as
$L = (J_{HCHO} + k_{HCHO}\,[OH])HCHO$.              (A3)
$D = -kdil\,(HCHO - HCHO_{bkg}) = -\frac{1}{\Delta CO}\frac{d\,\Delta CO}{dt}\,HCHO$ . 

457                                                                   (A4)

$P = \frac{d\,HCHO}{dt} + L + D = \Delta\,CO\,\frac{d\,nHCHO}{dt} + nHCHO\,\frac{d\,\Delta CO}{dt} + (J_{HCHO} + k_{HCHO}\,[OH])HCHO -$
$\frac{1}{\Delta CO}\frac{d\,\Delta CO}{dt}\,HCHO$ .              (A5)
By assuming HCHO >> HCHO bkg, $\frac{P}{\Delta CO}$ can be written as
$\frac{P}{\Delta CO} = \frac{dnHCHO}{dt} + (J_{HCHO} + k_{HCHO}[OH])nHCHO$.        (A6)
Where $\frac{d\,nHCHO}{dt}$ can be derived from measured HCHO and CO vs physical age; $J_{HCHO}$ is the HCHO
photolysis coefficient, derived from in-situ actinic flux measurements; OH is calculated from
VOCs ratios (Sect.2.4); $k_{HCHO}$ is the reaction rate coefficient of HCHO and OH.




Data and code availability:
Data are publicly available at https://www-air.larc.nasa.gov/missions/firex-aq/index.html with a
dataset doi: FIREX-AQ DOI: 10.5067/SUBORBITAL/FIREXAQ2019/DATA001. F0AM is
available at https://github.com/AirChem/F0AM. Model setup scripts for this study are available
from the contact author upon request.

Author contribution:
GMW and TFH directed the research direction. JL analyzed the data and discussed the results with
GMW. JL wrote the manuscript. TFH, GMW, JMS, JL, and RAH made ISAF HCHO
measurements. JBG, AL, and VS made iWAS measurements. GSD, JBN, HSH, JPG made
DACOM CO measurements. SRH and KU made CAFS photolysis frequencies measurements.
CDH, CHF, and AA provided the trajectories-based plume physical age. HSH provided MCE
calculation. TBR, JP, and IB made $O_3$ measurements. CW, MMC, GIG, and KS made PTR-ToF-
MS VOC measurements. AF, DR, and PW made CAMS HCHO measurements. ECA and RSH
made TOGA VOC measurements. SSB, CCW, MAR, and RAW made ACES measurements. PRV
and JAN made CIMS measurements. All authors reviewed and commented on the manuscript.

Acknowledgement:
We gratefully acknowledge the crew, logistical personnel, science team and science leadership
who facilitated the FIREX-AQ mission. We also thank Gao Chen and Ali Aknan for the merged
DC8 dataset used in this study. JL, GMW, RAH, JMS, and TFH acknowledge support from the
NASA Tropospheric Composition Program and NOAA Climate Program Office's Atmospheric
Chemistry, Carbon Cycle and Climate (AC4) program (NA17OAR4310004). SRH and KU were





funded by the NASA Tropospheric Composition Program (80NSSC18K0638). This material is
based upon work supported by the National Center for Atmospheric Research, which is a major
facility sponsored by the National Science Foundation under Cooperative Agreement No. 1852977.
KS acknowledges the support from the fund a Grant-in-Aid for Scientific Research (C) (18K05179)
from the Ministry of Education, Culture, Sports, Science and Technology of Japan.



Figures:



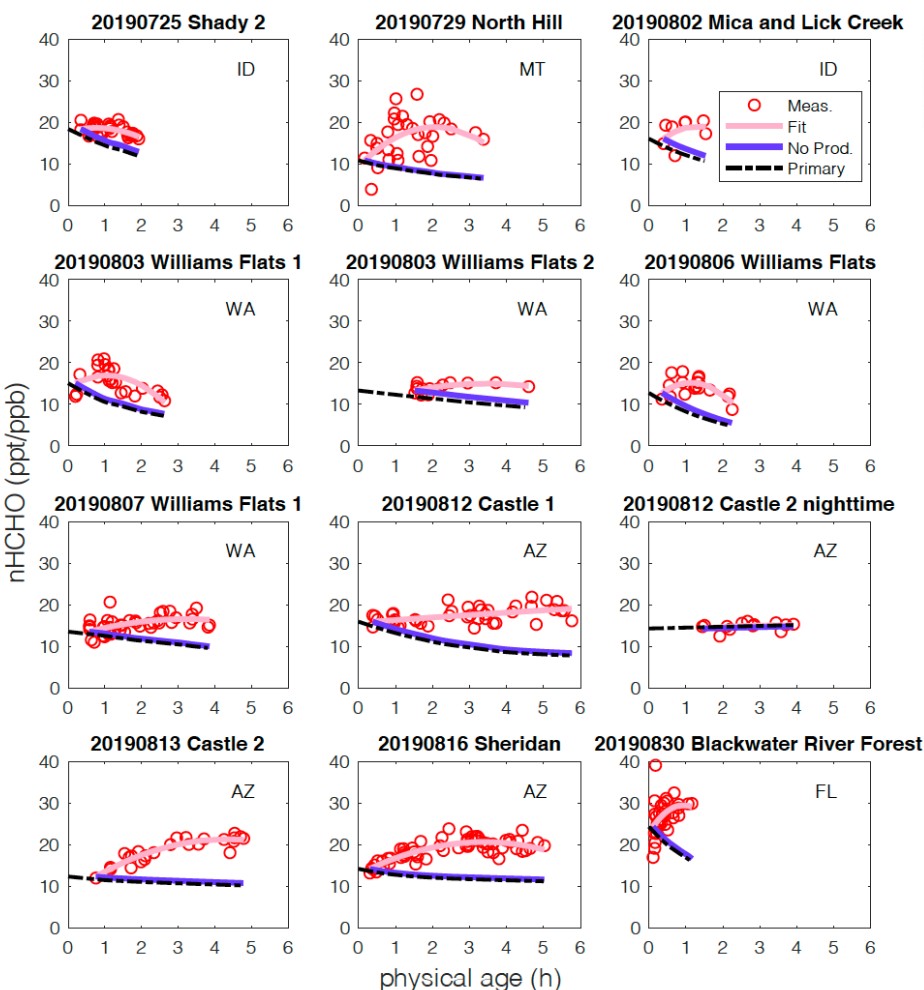


Figure 1. Observed nHCHO (HCHO to CO NEMR) trends (red circle), quadratic polynomial fit

(pink curve), calculated decay of nHCHO trend without secondary production (blue curve) using

measured photolysis rates, and calculated primary nHCHO trends (black dashed curve) with

physical age for the 12 wildfire plumes. The state of fire location for each fire plume is listed.

504

505



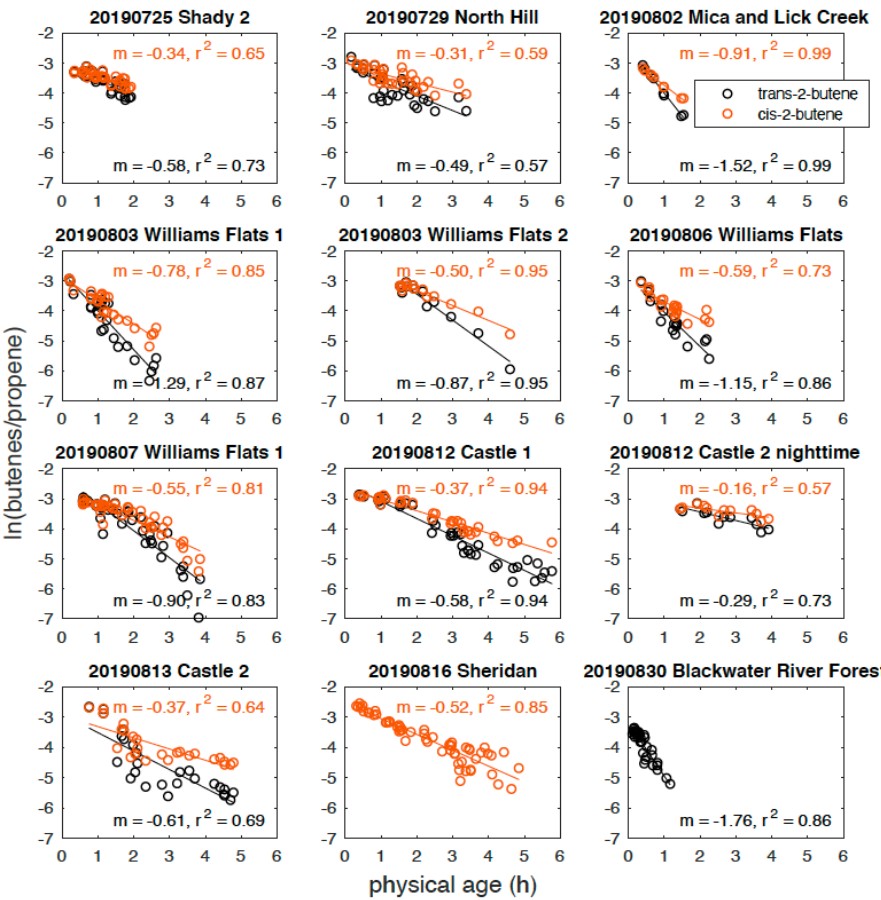

Figure 2. Natural logarithms of cis-2-butene to propene ratios (red circles) and trans-2-butene to propene ratios (black circles) vs. physical age for 12 wildfire plumes. The natural logarithms of the butenes/propene ratios correlate well with physical age with correlation coefficients, $r^2$, $\geq 0.57$. The slopes of the linear fits to the data (m, shown on the plots) reflect the oxidation by OH and $O_3$ and are used to calculate the average OH concentrations with average $O_3$ concentrations and reaction rate coefficients.

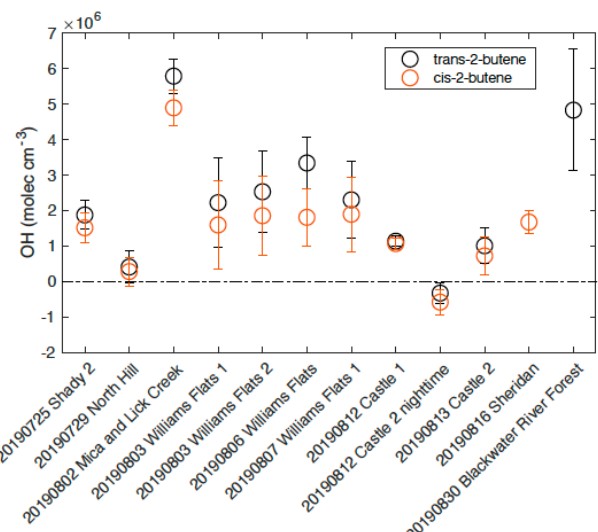


Figure 3. Estimated average OH concentrations for the plumes analyzed from the decay of trans-
2-butene – propene (black) and the decay of cis-2-butene – propene (red). The error bars represent
the propagated uncertainties from the slopes of butenes – propene decay and ozone variability
within the plume.

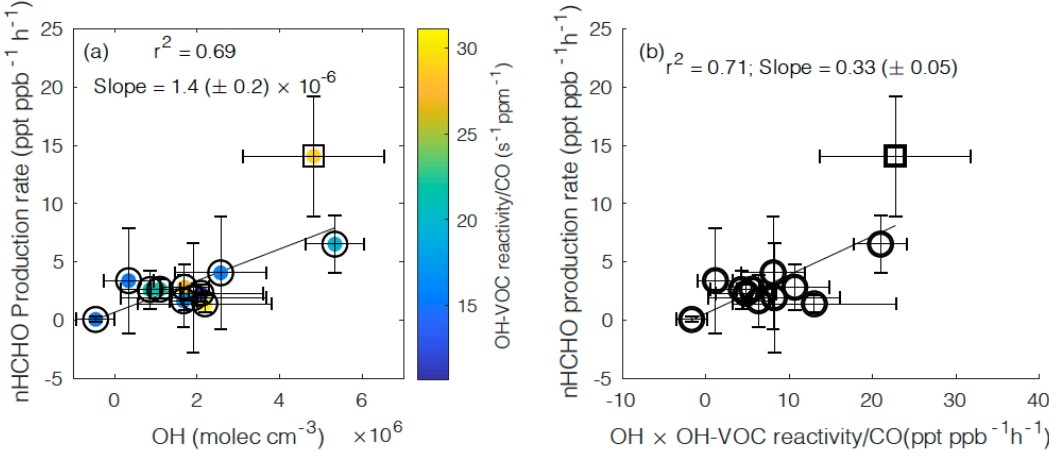


Figure 4. (a) Average secondary nHCHO production rate vs. average OH concentration, color-
coded by OH-VOC reactivity, for the 12 plumes including 11 western US wildfire plumes (circles)



and 1 eastern US wildfire plume (square). An uncertainty weighted linear York regression (Derek,
1968) yields a slope = 1.4 ($\pm$0.2)$\times$ $10^{-6}$ and $r^2$ = 0.69 for the 12 wildfire plumes.  (b) Average
secondary nHCHO production rate vs. the average product of OH and OH-VOC reactivity
normalized to CO (OH$\times$ OH-VOC reactivity/CO) for each plume. An uncertainty weighted linear
York regression yields a slope = 0.33 ($\pm$ 0.05) and $r^2$ = 0.71. The slope represents the estimated
effective yield $\alpha_{eff}$ of HCHO per VOC molecule oxidized by OH for the US wildfires.





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
