# Peer review of "Formaldehyde evolution in U.S. wildfire plumes during"

_Atmospheric Chemistry and Physics, 2021_

## Referee Comment (RC2)

**Review of "Formaldehyde evolution in U.S. wildfire plumes during FIREX-AQ"**

July 1, 2021

Liao et al. "Formaldehyde evolution in U.S. wildfire plumes during FIREX-AQ" uses data from $\sim 12$ fire plumes to study the relative importance of primary vs. secondary HCHO production in fire plumes along with the dominant factor controlling the magnitude of secondary HCHO production. The main conclusion is that OH concentrations, rather than total VOC reactivity with OH, is the main factor setting the magnitude of secondary HCHO production. The topic is appropriate for publication in ACP.

While the general methodology is sound as far as I can tell, I agree with the other reviewer that a more detailed explanation of how the authors arrived at their conclusions would strengthen this paper. In particular, I find that Fig. 4 in the main paper is not sufficient support of the primary conclusion (that OH concentrations rather than VOC reactivity are the primary control on HCHO production). There are also issues with definitions or concepts being introduced out of order that make the manuscript difficult to follow.

Thus, I recommend that revision be required to address these issues before final publication in ACP.

**Major comments**

- In Fig. 4a, as I understand it, the $y$-axis values of nHCHO production depend on plume OH concentration (in order to determine the loss rate of primary nHCHO to subtract primary from total nHCHO) and is plotted against OH concentration. This makes me concerned that the correlation might be driven in part by the relationship encoded in that calculation, rather than the physical relationship between OH and secondary HCHO production. Could you describe any tests you have done to determine whether that is the case?

- Also for Fig. 4 and the related text, I do not find it convincing to show secondary nHCHO production vs. OH concentrations colored by the VOC reactivity as evidence

that the former controls the nHCHO production rate. It is very difficult to evaluate the correlation between the $y$-axis and the colors. It would be a stronger argument, in my opinion, to have a panel that shows nHCHO production vs. OH-VOC reactivity (perhaps colored by OH concentration) to directly compare with the current panel (a). Further, I did not see anywhere that the $R^2$ value, slope, or slope uncertainty for the regression of nHCHO production vs. OH-VOC reactivity were reported. To claim that the correlation of nHCHO production with OH concentration is greater than the correlation of nHCHO production with OH-VOC reactivity without providing the latter value is rather unconvincing. Please provide this comparison.

- My last point on Fig. 4 is that $R^2$ values tend to be driven up by outliers. In Fig. 4, I suspect that the $R^2$ is being increased by one or both of the data points with [OH] $> 4 \times 10^6$ molec. cm$^{-3}$. I recommend the authors apply a bootstrap analysis to test the effect of these two points and calculate the uncertainty on the $R^2$ and confirm the uncertainty on the slope.

- In Sect. 3.3, lines 369 to 371, I am confused by the statement "Plume-average normalized OH-VOC reactivity...is lower than total OH reactivity across the analyzed plumes, and does not establish a clear relationship with OH."

  - It is not clear from the writing why there would be any expectation that OH-VOC reactivity would correlate with OH; one depends on VOC concentrations and rate constants, the other on OH itself. If the assumption is that increased OH would tend to decrease OH-VOC reactivity because the OH will react with the most reactive VOCs first, that is not explained in this section—and would also likely be a very non-linear relationship, so I would not naively expect this to have a clear correlation.

  - Please be clear about the difference between "(normalized) OH-VOC reactivity" and "total OH reactivity." The latter is not defined explicitly anywhere in the paper, and it should also be made clear whether both quantities are being normalized to $\Delta$ CO or not. Further, if the "total OH reactivity" is the OH-VOC reactivity plus OH reactivity with CO and NO$_2$, then the statement in lines 369 to 371 that OH-VOC reactivity is lower than total reactivity is true by definition, and not particularly useful.

  - Given that it is not clear what the difference between these quantities is, it is impossible to understand how this difference tells anything about the relative importance of sources vs. sinks of OH. In addition to clearly defining both quantities, please provide more detail on the logic for how this difference informs the relative OH source/sink importance.

**Minor/technical comments**

- The pink curve in Fig. 1 is really hard to see, a color with more contrast to the red would be more visible.

- How late are the nighttime plumes? I wouldn't expect there to be much OH chemistry after sunset with no photolysis, unless there is some longer-lived $HO_x$ reservoir in these plumes.

- Sect. 2.2 does not include justification for using $\Delta CO$ as a method to normalize for dilution. While I recognize that CO is often used in this manner, this section should include either (a) a citation to previous work showing that $\Delta CO$ is a reasonably accurate metric for dilution (ideally in biomass burning plumes) or (b) demonstrate using FIREX-AQ data that normalizing by $\Delta CO$ does account for dilution.

  - Since using CO as a tracer generally requires that production or loss of CO be minor relative to the concentration of CO in the plume (Müller et al. 2016), it would be helpful to show that this is true in these plumes. For example, while I assume that the amount of CO produced from $HCHO + OH \rightarrow HO_2 + CO$ ($\sim 3$ ppb/hr at 298 K) is small enough to not impact this analysis, without knowing the CO mixing ratios in the plumes, I cannot be sure.

- Please confirm that Eq. (3) is provided and used correctly. Following on from Eq. (2), I assume that the whole term multiplied by $t$ is set equal to the slope, thus:

$$slope_{butene} = (k_{butene+OH} - k_{propene+OH})[\text{OH}] + (k_{butene+O_3} - k_{propene+O_3})[\text{O}_3]$$
$$\Rightarrow slope_{butene} - (k_{butene+O_3} - k_{propene+O_3})[\text{O}_3] = (k_{butene+OH} - k_{propene+OH})[\text{OH}]$$
$$\Rightarrow \frac{slope_{butene} - (k_{butene+O_3} - k_{propene+O_3})[\text{O}_3]}{(k_{butene+OH} - k_{propene+OH})} = [\text{OH}]$$

Specifically, in Eq. (3) it looks like the sign of the $O_3$ term is wrong and the order of terms in the denominator is reversed.

**References**

Müller, Markus et al. (2016) "In situ measurements and modeling of reactive trace gases in a small biomass burning plume." *Atmos. Chem. Phys.*, 16, 3813–3824. doi: 10.5194/acp-16-3813-2016

---

## Author Comment (AC1)

The reviewers' comments are in black. Our responses to reviewers' comments are in blue and our modifications in the manuscript are in green.

Reviewer 1:

This manuscript analyzes formaldehyde data from select FIREX flights to better understand its production and loss and the drivers of secondary HCHO production. Overall, I found the science hard to follow as limited detail were given for the analyses. The descriptions and explanations need to be expanded to elevate the contribution and scientific the impact of the paper.

We will provide more detailed information as we address the following specific comments.

Specific Comments:

Line 192: How consistent was the background HCHO concentration? Was a single value appropriate to use as a cutoff?

The background HCHO concentrations did vary. We changed line 187-188 to "Wildfire plumes that meet the conditions listed below are above the background HCHO concentrations, which typically vary from 100 ppt – 1 ppb during FIREX-AQ, and are selected to study the evolution of HCHO in wildfires." The fire plume from Tucker on 0730 was the only plume filtered out according to condition (b). We found that the sampling pattern of this plume was also not really Lagrangian. Therefore, original condition (b) was deleted.

Line 210: cited the paper! Coggon, Matthew M., Christopher Y. Lim, Abigail R. Koss, Kanako Sekimoto, Bin Yuan, Jessica B. Gilman, David H. Hagan, et al. "OH Chemistry of Non-Methane Organic Gases (NMOGs) Emitted from Laboratory and Ambient Biomass Burning Smoke: Evaluating the Influence of Furans and Oxygenated Aromatics on Ozone and Secondary NMOG Formation." *Atmospheric Chemistry and Physics* 19, no. 23 (December 10, 2019): 14875–99. https://doi.org/10.5194/acp-19-14875-2019.

The paper is also cited now. Because using anhydride/furan ratio to estimate OH is not clearly covered by the referenced paper, communication notes are still included.

Line 222: Which did you use or did the combination of the two help constrain the uncertainty? The slopes are different for each compound – that is due to the different k values of OH and O3 for the 2 compounds? The use of these two compounds should be more explicitly described with more specifics about what the different slopes indicate. Figure 2 or S2 (how are they different?) aren't that helpful to your discussion. Yes all the slopes look pretty good - should we take away more than that?

Figure S2 is modified to include the eastern US wildfire plume. Because now fig. S2 included all the plots in original fig. 2, original fig. 2 is now removed.

[Figure]

Figure S2. Natural logarithms of cis-2-butene to propene ratios (red circles) and trans-2-butene to propene ratios (black circles) vs. physical age for 18 western US and 1 eastern US wildfire plumes that met selection conditions a) in Sect. 2.3. 25 July Shady 3 plume was not plotted because of the unavailability of iWAS data. The plumes with good correlations ($r^2 \geq 0.57$) between natural logarithms of the butenes/propene ratios and physical age and with sufficient data (data points > 8) are selected for this analysis. The slopes of the linear fits to the data (m, shown on the plots) reflect the oxidation by OH and $O_3$ and are used to calculate the average OH concentrations with average $O_3$ concentrations and reaction coefficients.

Line 222-224 changed to "Both cis-2-butene/propene ratio and trans-2-butene/propene ratio are used to estimate OH because these gases have lifetimes comparable to physical age (2–6 h) for most of the analyzed plumes and both OH from the two VOC ratios are averaged to calculate the estimated OH." As shown in fig. 1 (now), OH from each is not statistically different in most of the plumes.

Line 229 added "Different slopes between cis-2-butene/propene and trans-2-butene/propene vs plume age (Fig. S2) depend on the difference in reaction rate coefficients of OH and $O_3$ with 2-butene (cis-2-butene and trans-2-butene) and propene, in addition to OH and $O_3$ concentrations, as shown in eqn (2)."

Line 344 added "Figure S2 includes all the plumes that meet selection condition (a) in sect.2.3 and 12 plumes with good correlations ($r^2 = 0.57$–$0.99$) between 2-butenes/propene and plume age and sufficient data (data points > 8) are selected for this analysis. The slopes in Fig. S2 infer the estimated OH concentrations and their coefficients of determination ($r^2$) imply how good the VOC decay can be used to estimate OH."

Line 230: Including the k values you used here and whether they were corrected for the ambient temperature is important.

Line 229-230 change to "These reaction rate coefficients are those reported by Atkinson et al. (2006) with real time temperature and pressure dependence. The plume average reaction rate coefficients are $k_{propene\_OH} = 3.1 \times 10^{-11}$ $cm^3$ $molec^{-1}s^{-1}$, $k_{cis-2-butene\_OH} = 6.4 \times 10^{-11}$ $cm^3$ $molec^{-1}s^{-1}$, $k_{trans-2-butene\_OH} = 8.0 \times 10^{-11}$ $cm^3$ $molec^{-1}s^{-1}$, $k_{propene\_O_3} = 6.4 \times 10^{-18}$ $cm^3$ $molec^{-1}s^{-1}$, $k_{cis-2-butene\_O_3} = 9.9 \times 10^{-17}$ $cm^3$ $molec^{-1}s^{-1}$, and $k_{trans-2-butene\_O_3} = 1.5 \times 10^{-16}$ $cm^3$ $molec^{-1}s^{-1}$."

Line 235: Again which butane compound are you using for your OH calculation? Both? [This becomes apparent later but this is where I want to know the details]

Line 239 added "Butene in eqn (2) and (3) represents trans-2-butene or cis-2-butene, both of which are used in average OH estimation."

Line 236: Often there are O3 deficits in the smoke plume center due to the rapid chemistry happening creating strong gradients in O3 concentration. How sensitive are your derived OH concentrations to the range of O3 in a particular transect? I realize that you say the uncertainty of O3 variation is taken into account in the total plume-average OH uncertainty but on a component by component basis how much uncertainty is each term contributing? Also reference your table in the supplement here with the OH uncertainty and add the OH concentration to the table too so we can compare the uncertainty to the value.

Line 235 changed to "Because the instantaneous $O_3$ measurements do not reflect the oxidation history, the average ozone concentration of the entire circuit with multiple transects is used to represent the integrated $O_3$ effect on alkene oxidation."

Line 239 added "$O_3$ variation, uncertainty in OH due to $O_3$ variation, total OH uncertainty and estimated OH are listed in Table S4."

Table S4. Mean and standard deviation of $O_3$ concentrations, OH uncertainty due to $O_3$ variation, total OH uncertainty, and estimated OH of the 12 plumes

| Plume sampling date | $O_3$ mixing ratios (mean±std, ppb) | OH uncertainty due to $O_3$ variability $\times 10^6$ (molecules cm$^{-3}$) | Total OH uncertainty $\times 10^6$ (molecules cm$^{-3}$) | Estimated OH $\times 10^6$ (molecules cm$^{-3}$) |
|---|---|---|---|---|
| 20190725 | 32.0±5.7 | 0.31 | 0.59 | 1.69 |
| 20190729 | 51.2±1.6 | 0.15 | 0.60 | 0.34 |
| 20190802 | 55.5±6.7 | 0.51 | 0.70 | 5.34 |
| 20190803 | 88.2±18.6 | 1.51 | 1.76 | 1.90 |
| 20190803 | 43.7±19.2 | 1.55 | 1.62 | 2.19 |
| 20190806 | 58.3±4.3 | 0.36 | 1.10 | 2.57 |
| 20190807 | 60.4±23.5 | 1.42 | 1.50 | 2.09 |
| 20190812 | 50.6±2.3 | 0.14 | 0.23 | 1.10 |
| 20190812nighttime | 47.5±0.8 | 0.05 | 0.46 | -0.45 |
| 20190813 | 56.1±4.4 | 0.26 | 0.72 | 0.86 |
| 20190816 | 63.1±6.5 | 0.34 | 0.33 | 1.67 |
| 20190830 | 74.4±17.3 | 2.04 | 1.71 | 4.83 |

Line 252: 27% higher is not slight but from the figure it does appear to be within the error of the calculation. Either say that or do a hypothesis test to show they aren't statistically different. Explain why there might be a systematic bias in the reaction rate at low temperature. Do you see a trend in the comparison with temperature? Is there a study you can cite to support the suspected bias in k?

I think this comment is more relevant to line 352 instead of line 252.

Line 350-354 changed to "Plume-to-plume variability in average OH concentrations is generally consistent between the two ratio methods. OH concentrations from trans-2-butene are systematically higher than those from cis-2-butene by 27% on average, which may reflect systematic bias in reaction rate coefficients or observations. For all plumes where both calculations were available, differences are within the combined uncertainties."

Line 305: I suggest showing the OH concentrations first (Fig 3) since they are used in Figure 1 for the blue curves. When they are first mentioned is when I want to know more about them.

As suggested, we moved OH concentrations section to section 3.1.

Line 309: Does it really represent an upper limit on the emitted HCHO? That implies you know that there was no loss of HCHO in the plume prior to measurement. What evidence do you have to support this?

Line 309-311 changed to "The calculated nHCHO without production is higher than primary (emitted) nHCHO because some HCHO production and loss had already occurred before the closest transect.

Line 313: What is the difference between the blue and the black curves? Blue: predicted decay of primary nHCHO from J and OH. Black: calculated primary nHCHO. These seem to be almost the same definition - or is there very little loss of the primary nHCHO. Perhaps refer to the equations to highlight which terms are different? The two lines are pretty similar for all shown cases – do both need to be shown? What is the main goal of showing both of these calculated trends? It would be better to show a figure related to the discussion of the fraction of primary v secondary HCHO over the lifetime of the plume (as discussed in the text) and how it varies. A figure like I just described would facilitate your analysis of the drivers of HCHO.

Thanks for the comment. The black curves are deleted as they are similar to the blue curves in fig. 2.

[Figure]

Figure 2. Observed nHCHO (HCHO to CO NEMR) trends (red circle), quadratic polynomial fit (pink curve), and calculated decay of nHCHO trend without secondary production (blue curve) using measured photolysis rates along plume physical age for the 12 wildfire plumes. The state of fire location for each fire plume is listed.

A figure of the fraction of primary and secondary HCHO for these plumes vs. plume physical age are shown in Fig. S8.

[Figure]

Fig. S8. Fraction of primary and secondary nHCHO vs. plume physical age for the 12 plumes. The fraction of primary nHCHO is estimated by assuming nHCHO and the loss rate of nHCHO were constant between emission and the closest observation. The slight increase in primary nHCHO fraction with physical age for the 20190803 Williams Flats 1 may be due to the uncertainty in the polynomial fit of the observed nHCHO, the nHCHO loss rate calculation, Lagrangian plume assumption, or emission variation.

Line 313-316 changed to "The fraction of primary and secondary nHCHO varies from plume to plume and depends on secondary HCHO production rates and total HCHO loss rates. This can be inferred from nHCHO trends and the loss-only nHCHO decays in Fig. 2 and is also shown in Fig. S8. We estimate the fraction of primary HCHO by assuming nHCHO and the loss rate of nHCHO are constant between emission and the closest observation."

Line 319 added "The slowing down increase in secondary nHCHO fraction with time probably indicate that the production of secondary nHCHO slows down with physical age."

Line 322: This is not obvious in the figure since most start out with a positive trend in the measured values with time and then the loss overtakes the production. It just happens faster in the 3 you highlight with larger loss rates than production in the table. Perhaps you can color the fit line to show if the plume net loss exceeded production to make it clearer? It might be more informative to show the role of J and OH loss and the balance with production across the physical ages of the plume. A figure like this would more clearly show the point that I believe you are trying to make (what are the controls on HCHO concentration in fire plumes and how do they vary).

Instead of changing the fit curves colors, we added more text as below.

Line 323-324 changed to "Plumes exhibiting overall negative nHCHO trends (0725 Shady 2, 0803 Williams Flats 1 and 0806 Williams Flats in Fig. 3) have higher nHCHO loss rates than production rates (Table S3)"

Added a plot nHCHO production rate, loss rate and change rate vs. physical age in the SI as Fig S10.

Fig S10

[Figure]

Fig. S10 nHCHO production (gold), loss (blue) and change (red) rates with physical age for the 12 plumes. The uncertainty (red error bars) in nHCHO change rate is estimated from the difference between measured nHCHO and the polynomial fit. The uncertainty (blue error bars) in nHCHO loss rate is estimated from the uncertainty in OH estimation and the difference between the loss rate calculated from the measured photolysis rates and temperature dependent reaction rate coefficient and the loss rate calculated from the interpolation of the average photolysis rates and reaction rate coefficient. The uncertainty (gold error bars) in nHCHO production rate is the combined uncertainty in nHCHO loss rate and change rate. The uncertainty accounts for the majority of the negative calculated nHCHO production rates. The negative nHCHO production rate at the end of the 0803 Williams Flats 1 plume cannot be not fully accounted by the estimated uncertainty. This may be due to emission variation or uncertainties in the lagrangian plume sampling assumption for air masses downwind away from the source.

Line 335 added "Besides the variability among different plumes, nHCHO production and loss also vary within a plume across physical age. In all analyzed plumes, the nHCHO slope shifts from positive to neutral or negative within the first 2-6 hs (Fig. 2). Figure S10 shows the age progression of nHCHO production, loss, and net change for the 12 plumes. In general, both production and loss decrease with age. Decreases in both are expected due to declining solar radiation, which results from the typical late-afternoon FIREX-AQ sampling strategy. Reduced production with increasing age may also reflect the decay of reactive VOC and oxidant (e.g., HONO) precursors."

Section 3.2: This section needs to be the first part of the results and discussion section since the OH concentration is used to calculate the loss of primary nHCHO.

Revised per the reviewer's suggestion.

Section 3.3: At the beginning of this section remind the reader how you are determining secondary HCHO production - a mass balance approach with loss, production, and dilution terms - and not from VOC chemistry.

Line 360 added "The secondary nHCHO production rate is determined by a mass balance approach with loss, production, and dilution terms, as discussed in Sect. 2.5."

Line 360: Since secondary production was calculated with the OH concentrations I would expect there to be a correlation between the 2 terms. How does the correlation with J compare to that with OH? Or other oxidants? A more comprehensive analysis and discussion would guide the reader to the same conclusion that you make.

Line 362 added "Although OH concentrations are used to calculate secondary nHCHO production rates, the nHCHO loss term ($k$[OH]nHCHO) due to OH only accounts for 2-35% of all the terms on the righthand side of eqn(6), which is used to calculate secondary nHCHO production rate for the plumes. This indicates that the good correlation between the secondary nHCHO production rate and OH is not due to the inclusion of OH in the nHCHO production rate calculation. The nHCHO secondary production rates also correlate with the HCHO photolysis rates ($r^2 = 0.53$ uncertainty weighted linear regression), which is not unexpected as OH and $J_{HCHO}$ positively

correlate as well. The correlation between nHCHO secondary production rates with oxidant ozone is poor ($r^2 = 0.1$ from bivariate regression as uncertainty weighted linear regression does not yield a reasonable fit)."

Line 366: A high R2 doesn't necessarily mean that the relationship is significant. Including a statistical analysis with the p values with strengthen the conclusions you are making.

*P-values are added to Fig. 3 both panels.*

Line 371 added "*P*-values in Fig. 3 show the correlation between nHCHO production rate vs. OH or vs. OH× normalized OH-VOC reactivity is statistically significant ($p < 0.05$)"

Line 368: What other potential drives of secondary HCHO production did you look at? How does the trend/relationship change if the eastern US fire is excluded? It looks pretty different (high VOCs and nHCHO) and there is only one fire from that region.

As mentioned above, line 362 added "The nHCHO secondary production rates also correlate with the HCHO photolysis rates ($r^2 = 0.53$ uncertainty weighted linear regression), which is not unexpected as OH and $J_{HCHO}$ positively correlate as well. The correlation between nHCHO secondary production rates with oxidant ozone is poor ($r^2 = 0.1$ from bivariate regression as uncertainty weighted linear regression does not yield a reasonable fit)."

Line 368 added "Although there is only one eastern US wildfire plume sampled during FIREX-AQ, it has high VOCs, nHCHO, nHCHO production rate, and OH, and the inclusion of the eastern US wildfire increases the coefficient of determination ($r^2$ from 0.54 to 0.69) and the slope (m from 0.30 to 0.33) of nHCHO secondary production rates vs. OH. More wildfire sampling is needed to understand the difference between western and eastern US wildfires."

Line 370: Why exclude NO2 and CO in the OH reactivity analysis? If interested in the role of VOCs I understand but the controls on OH concentration will still include NO2 and CO. I more complete analysis looking at both the total and VOC reactivity would improve the work since I expect there is variability in the NO2 (and CO) that makes the VOC/total reactivity vary by plume.

Line 371 added "Because the yield of HCHO from VOC oxidation is calculated in the study, normalized OH-VOC reactivity instead of normalized total OH reactivity is mainly used. A plot of nHCHO production rate vs normalized total OH reactivity color coded with OH is shown in Fig. S11b. Similar to Fig. S11a, the correlation between nHCHO production rate with normalized total OH reactivity is also not significant."

[Figure]

Figure S11 (a). Average nHCHO production rate vs. normalized OH-VOC reactivity (OH-VOC reactivity/CO) for the 12 plumes including 11 western US wildfire plumes (circles) and 1 eastern US wildfire plume (square). Unweighted bivariate linear regression was applied to fit the data because uncertainty weighted linear regression does not yield a reasonable fit. The unweighted (or equally weighted) bivariate linear regression yields a slope = 0.31, $r^2$ = 0.14, and $p$ = 0.2 for the 12 wildfire plumes. (b) Average secondary nHCHO production rate vs. normalized total OH reactivity (total OH reactivity/CO) for the 12 plumes including 11 western US wildfire plumes (circles) and 1 eastern US wildfire plume (square). An unweighted (or equally weighted) bivariate linear regression yields a slope = 0.32, $r^2$ = 0.22, and $p$ = 0.1 for the 12 wildfire plumes.

The contribution of CO and $NO_2$ to OH reactivity is stated in Line 369 to 371 "Plume-average normalized OH-VOC reactivity ranges from 11 to 31 s[-1] (ppm CO)[-1], which is about 20% lower than normalized total OH reactivity across the analyzed plumes."

Line 372: This sentence is a repeat of 368 and it still isn't clear if you actually are showing this.

Line 372 deleted "This demonstrates that variability in OH, as well as secondary nHCHO production, likely depends principally on variability in OH sources (e.g., photolysis of HONO and conversion of $HO_2$ by NO) (Peng et al., 2020) rather than sinks."

Line 380: I don't understand this logic since it seems to contradict your analysis in the previous paragraph where you said a strong correlation indicated OH was an important driver. I can't really tell how different [OH-VOC reactivity/CO] is from [OH-VOC reactivity/CO * OH] but I imagine the valves are scaled pretty linearly. It might be more informative [OH-VOC reactivity/CO] on the x-axis and getting rid of the colors in 4a since they are hard to see anyway. You could also show with [OH-VOC reactivity/CO * OH] to get the effective yield.

As the reviewer suggested, nHCHO production rate vs. OH-VOC reactivity/CO is added to Fig. S11 (a) to show that plume-to-plume variability in reactive VOC availability is a comparatively minor driver of nHCHO production.

See Figure S11 above.

We agree with the reviewer that the colors in Fig. 3a (originally Fig. 4a) are not clear to see. Fig.3a is modified to show the colors upfront. The plot of nHCHO production rate vs OH-VOC reactivity/CO × OH has been shown in Fig. 3b (originally Fig. 4b).

[Figure]

Figure 3. (a) Average secondary nHCHO production rate vs. average OH concentration, color-coded by normalized OH-VOC reactivity, for the 12 plumes including 11 western US wildfire plumes (circles) and 1 eastern US wildfire plume (square). An uncertainty weighted linear York regression (Derek, 1968) yields a slope = 1.4 ($\pm$0.2)$\times$ $10^{-6}$ and $r^2$ = 0.69 ($\pm$0.16) for the 12 wildfire plumes. (b) Average secondary nHCHO production rate vs. the average product of OH and OH-VOC reactivity normalized to CO (OH$\times$ OH-VOC reactivity/CO) for each plume. An uncertainty weighted linear York regression yields a slope = 0.33 ($\pm$ 0.05) and $r^2$ = 0.71 ($\pm$0.19). The slope represents the estimated effective yield $\alpha_{eff}$ of HCHO per VOC molecule oxidized by OH for the US wildfires. The uncertainties in $r^2$ are from bootstrap analysis. $P$ value in each panel is to evaluate if linear correlation is statistically significant ($p<$ 0.05).

Line 401: What were the PTRMS measurements used for? It is unclear as written and how this information is related to the current discussion. Why is this one compound so important?

Line 401 to 405 changed to "Species that are highly reactive and present in large quantities such as $CH_3CHO$ are important for OH-VOC reactivity and $\alpha_{eff}$ calculation. We use PTRMS $CH_3CHO$ in the OH-VOC reactivity calculation because they are more easily integrated over the iWAS sampling time than the TOGA $CH_3CHO$.".

Line 407-11: You should not be comparing western and eastern fires give the number of eastern fires in this analysis is 1. You have no idea what if the fire was representative of other fires in the region. I suggest rewriting/adding that more data from eastern fires are needed to understand how they may be different as suggested by this one fire.

Line 407-408 changed to "The $\alpha_{eff}$ for the one eastern US wildfire plume is higher than for the western US wildfire plumes but more sampling of eastern wildfire plumes is needed to determine if there is a statistical difference in $\alpha_{eff}$."

Line 436: Did you show that the variability in the reactive VOC pool is not playing an important role? I'm not sure the analysis presented before this does a good job of this since the figure is weighted by OH.

A panel about nHCHO production rate vs. OH-VOC reactivity/CO is added to fig.S11a to show the reactive VOC pool is not playing an important role as OH.

Line 372 added "nHCHO production rates vs. normalized OH-VOC reactivity (Fig. S11a) shows a lower coefficient of determination ($r^2$) and a higher $p$ value than Fig.3a."

No technical comments.

---

## Author Comment (AC2)

The reviewers' comments are in black. Our responses to reviewers' comments are in blue and our modifications in the manuscript are in green.

Reviewer 2

Liao et al. Formaldehyde evolution in U.S. wildfire plumes during FIREX-AQ" uses data from ~12 fire plumes to study the relative importance of primary vs. secondary HCHO production in fire plumes along with the dominant factor controlling the magnitude of secondary HCHO production. The main conclusion is that OH concentrations, rather than total VOC reactivity with OH, is the main factor setting the magnitude of secondary HCHO production. The topic is appropriate for publication in ACP.
While the general methodology is sound as far as I can tell, I agree with the other reviewer that a more detailed explanation of how the authors arrived at their conclusions would strengthen this paper. In particular, I find that Fig. 4 in the main paper is not sufficient support of the primary conclusion (that OH concentrations rather than VOC reactivity are the primary control on HCHO production). There are also issues with definitions or concepts being introduced out of order that make the manuscript difficult to follow.
Thus, I recommend that revision be required to address these issues before final publication in ACP.

To address the reviewer's concern that Fig. 4 in the main paper is not sufficient support of the primary conclusion (that OH concentrations rather than VOC reactivity are the primary control on HCHO production), Fig. 3a (originally Fig. 4a) is modified to show the colors upfront so that it is clear to see the dependence of nHCHO production rate on normalized OH-VOC reactivity. In addition, the scatter plots of nHCHO production rate vs. normalized VOC-OH reactivity and vs. normalized total OH reactivity are added in fig. S11a and fig. S11b.

[Figure]

Figure 3. (a) Average secondary nHCHO production rate vs. average OH concentration, color-coded by normalized OH-VOC reactivity, for the 12 plumes including 11 western US wildfire plumes (circles) and 1 eastern US wildfire plume (square). An uncertainty weighted linear York regression (Derek, 1968) yields a slope = 1.4 ($\pm$0.2)$\times 10^{-6}$ and $r^2$ = 0.69 ($\pm$0.16) for the 12 wildfire plumes. (b) Average secondary nHCHO production rate vs. the average product of OH and OH-VOC reactivity normalized to CO (OH$\times$ OH-VOC reactivity/CO) for each plume. An uncertainty weighted linear York regression yields a slope = 0.33 ($\pm$ 0.05) and $r^2$ = 0.71 ($\pm$0.19). The slope represents the estimated effective yield $\alpha_{eff}$ of HCHO per VOC molecule oxidized by OH for the US wildfires. The uncertainties in $r^2$ are from bootstrap analysis. P value in each panel is to evaluate if linear correlation is statistically significant ($p<$ 0.05).

[Figure]

Figure S11 (a). Average nHCHO production rate vs. normalized OH-VOC reactivity (OH-VOC reactivity/CO) for the 12 plumes including 11 western US wildfire plumes (circles) and 1 eastern US wildfire plume (square). Unweighted bivariate linear regression was applied to fit the data because uncertainty weighted linear regression does not yield a reasonable fit. The unweighted (or equally weighted) bivariate linear regression yields a slope = 0.31, $r^2$ = 0.14, and $p$ = 0.2 for the 12 wildfire plumes. (b) Average secondary nHCHO production rate vs. normalized total OH reactivity (total OH reactivity/CO) for the 12 plumes including 11 western US wildfire plumes (circles) and 1 eastern US wildfire plume (square). An unweighted (or equally weighted) bivariate linear regression yields a slope = 0.32, $r^2$ = 0.22, and $p$ = 0.1 for the 12 wildfire plumes.

**Major comments**

• In Fig. 4a, as I understand it, the y-axis values of nHCHO production depend on plume OH concentration (in order to determine the loss rate of primary nHCHO to subtract primary from total nHCHO) and is plotted against OH concentration. This makes me concerned that the correlation might be driven in part by the relationship encoded in that calculation, rather than the physical relationship between OH and secondary HCHO production. Could you describe any tests you have done to determine whether that is the case?

Line 362 added "Although OH concentrations are used to calculate secondary nHCHO production rate, the nHCHO loss term ($k$[OH]nHCHO) due to OH only accounts for 2-35% of all the terms on the righthand side of eqn (6), which is used to calculate secondary nHCHO production rate for the plumes. This indicates that the good correlation between secondary nHCHO production rate and OH is not due to the inclusion of OH in nHCHO production rate calculation."

• Also for Fig. 4 and the related text, I do not find it convincing to show secondary nHCHO production vs. OH concentrations colored by the VOC reactivity as evidence that the former controls the nHCHO production rate. It is very difficult to evaluate the correlation between the y-axis and the colors. It would be a stronger argument, in my opinion, to have a panel that shows nHCHO production vs. OH-VOC reactivity (perhaps colored by OH concentration) to directly compare with the current panel (a). Further, I did not see anywhere that the $R_2$ value, slope, or slope uncertainty for the regression of nHCHO production vs. OH-VOC reactivity were reported. To claim that the correlation of nHCHO production with OH concentration is greater than the correlation of nHCHO production with OH-VOC reactivity without providing the latter value is rather unconvincing. Please provide this comparison.

As suggested by the reviewer, a panel about nHCHO production rate vs. OH-VOC reactivity/CO with coefficient of determination and $p$ value is added to figure S11a.

Line 372 added "nHCHO production rates vs. normalized OH-VOC reactivity (Fig. S11a) shows a lower coefficient of determination ($r^2$) and a higher $p$ value than Fig.3a. Because uncertainty weighted linear regression does not yield a meaningful fit for Fig. S11a, unweighted (or equally weighted) bivariate linear regression is used."

• My last point on Fig. 4 is that $R_2$ values tend to be driven up by outliers. In Fig. 4, I suspect that the $R_2$ is being increased by one or both of the data points with [OH] > 4 *$10^6$ molec. cm$^{-3}$. I recommend the authors apply a bootstrap analysis to test the effect of these two points and calculate the uncertainty on the $R^2$ and confirm the uncertainty on the slope.

Bootstrap analysis is used to estimate the uncertainties in both the coefficients of determination $r^2$ provided in Fig. 3.

• In Sect. 3.3, lines 369 to 371, I am confused by the statement "Plume-average normalized OH-VOC reactivity...is lower than total OH reactivity across the analyzed plumes, and does not establish a clear relationship with OH."

--It is not clear from the writing why there would be any expectation that OH-VOC reactivity would correlate with OH; one depends on VOC concentrations and rate constants, the other on OH itself. If the assumption is that increased OH would tend to decrease OH-VOC reactivity because the OH will react with the most

reactive VOCs first, that is not explained in this section and would also likely be a very non-linear relationship, so I would not naively expect this to have a clear correlation.

Line 371 deleted ", and does not exhibit a clear relationship with OH"

--Please be clear about the difference between "(normalized) OH-VOC reactivity" and "total OH reactivity." The latter is not defined explicitly anywhere in the paper, and it should also be made clear whether both quantities are being normalized to CO or not. Further, if the "total OH reactivity" is the OH-VOC reactivity plus OH reactivity with CO and $NO_2$, then the statement in lines 369 to 371 that OH-VOC reactivity is lower than total reactivity is true by definition, and not particularly useful.

Line 370-371 changed "total OH reactivity" to "normalized total OH reactivity".

We agreed that normalized OH-VOC reactivity is lower than normalized total OH reactivity by definition, but the differences can vary. We pointed out that the normalized OH-VOC reactivity is 20% lower, which can vary from cases to cases, than the normalized total OH reactivity for these plumes. This is reflected in line 370-371:
"Plume-average normalized OH-VOC reactivity ranges from 11 to 31 $s^{-1}$ $(ppm\ CO)^{-1}$, which is about 20% lower than normalized total OH reactivity across the analyzed plumes."

In section 2.7, changed to "We calculate the OH-VOC reactivity ($\sum k_i\ VOC_i$) by excluding OH reactions with $NO_2$ and CO from the total OH reactivity and define normalized OH-VOC reactivity or normalized total OH reactivity as OH-VOC reactivity normalized by CO or total OH reactivity normalized by CO."

--Given that it is not clear what the difference between these quantities is, it is impossible to understand how this difference tells anything about the relative importance of sources vs. sinks of OH. In addition to clearly defining both quantities, please provide more detail on the logic for how this difference informs the relative OH source/sink importance.

Line 371-274 deleted "This demonstrates that variability in OH, as well as secondary nHCHO production, likely depends principally on variability in OH sources (e.g., photolysis of HONO and conversion of $HO_2$ by NO) (Peng et al., 2020) rather than sinks."

**Minor/technical comments**
• The pink curve in Fig. 1 is really hard to see, a color with more contrast to the red would be more visible.

Changed the pink curves to gray curves in Fig. 2 (originally Fig. 1).

[Figure]

Figure 2. Observed nHCHO (HCHO to CO NEMR) trends (red circle), quadratic polynomial fit (pink curve), and calculated decay of nHCHO trend without secondary production (blue curve) using measured photolysis rates along plume physical age for the 12 wildfire plumes. The state of fire location for each fire plume is listed.

• How late are the nighttime plumes? I wouldn't expect there to be much OH chemistry after sunset with no photolysis, unless there is some longer-lived $HO_x$ reservoir in these plumes.

Line 215 added "The nighttime plume on 12 August was after 8:00 pm local time with average $O_3$ photolysis rate of essentially zero."

• Sect. 2.2 does not include justification for using CO as a method to normalize for dilution. While I recognize that CO is often used in this manner, this section should include either (a) a citation to previous work showing that CO is a reasonably accurate metric for dilution (ideally in biomass burning plumes) or (b) demonstrate using FIREX-AQ data that normalizing by CO does account for dilution.

--Since using CO as a tracer generally requires that production or loss of CO be minor relative to the concentration of CO in the plume (Muller et al. 2016), it would be helpful to show that this is true in these plumes. For example, while I assume that the amount of CO produced from HCHO + OH → HO$_2$ + CO (3 ppb/hr at 298 K) is small enough to not impact this analysis, without knowing the CO mixing ratios in the plumes, I cannot be sure.

Line 166 added "Because photochemical production of CO is very small compared to the high CO concentrations in the biomass burning plumes (e.g., CO production from HCHO photolysis and OH oxidation for 1 hr is < 1% of CO concentrations in the plumes), trace gases concentrations are normalized to CO in the biomass burning plumes to account for dilution, as in many previous biomass burning studies (e.g., Müller et al., 2016; Selimovic et al., 2018)."

• Please confirm that Eq. (3) is provided and used correctly. Following on from Eq. (2), I assume that the whole term multiplied by t is set equal to the slope, thus:

slope$_{butene}$ = (k$_{butene+OH}$ -k$_{propene+OH}$)[OH] + (k$_{butene+O3}$ -k$_{propene+O3}$)[O$_3$]

$\Rightarrow$ slope$_{butene}$ - (k$_{butene+O3}$ - k$_{propene+O3}$)[O$_3$] = (k$_{butene+OH}$ - k$_{propene+OH}$)[OH]

$\Rightarrow \dfrac{Slope\ butene-(k_{butene+O3}-k_{propene+O3})[O3]}{k_{butene+OH}-k_{propene+OH}}$ = [OH]

Specifically, in Eq. (3) it looks like the sign of the O$_3$ term is wrong and the order of terms in the denominator is reversed.

Slope$_{butene}$ = $-\{(k_{butene+OH}-k_{propene+OH})[OH] +(k_{butene+O_3}-k_{propene+O_3})[O_3]\}$. The slope in your calculation is missing a minus sign.

Slope$_{butene}$ $+(k_{butene+O_3}-k_{propene+O_3})[O_3]= -(k_{butene+OH}-k_{propene+OH})[OH]$

[OH]=$\dfrac{slope_{butene} + (k_{butene+O_3}-k_{propene+O_3})[O_3]}{k_{propene+OH}-k_{butene+OH}}$.

**References**

Muller, Markus et al. (2016) "In situ measurements and modeling of reactive trace gases in a small biomass burning plume." Atmos. Chem. Phys., 16, 3813-3824. doi: 10.5194/acp-16-3813-2016.

---

## Author Response (AR2)

Thanks for addressing most of the reviewer concerns. Please address a remaining issues raised by reviewer "Thank you to the authors for addressing my concerns. I now recommend acceptance, but suggest three small additions. First, in the statement "e.g., CO production from HCHO photolysis and oxidation for 1 h is < 1% of CO concentrations in the plumes" please support this with a calculation, figure, citation, or other proof rather than a simple assertion. Second, please define what is meant by "reasonable fit" on line 354 and "meaningful fit" on line 366, as these are used to justify comparing two different types of fits but not given a standardized definition. Third, please include the bootstrapped uncertainty in R2 for Fig. S11 as done in Fig 3 so that the two figures can be fully compared with one another." in you revision. Thanks. Dubey

1. Changed to : e.g., CO production from HCHO photolysis and oxidation for 1 h is about 2.5 ppbv, which is < 1% of CO concentrations of 985 ppbv on average in the plumes.

2. Changed to "Because uncertainty weighted linear regression yields a low $r^2 = 0.08$ for Fig. S11a, unweighted (or equally weighted) bivariate linear regression is used."

   Changed to "$r^2 = 0.1$ from bivariate regression as uncertainty weighted linear regression does not yield a reasonable fit" to "$r^2 = 0.1$ from bivariate regression"

3. Bootstrapped uncertainties are added to Fig. S11.

[Figure]

Figure S11. (a). Average nHCHO production rate vs. normalized OH-VOC reactivity (OH-VOC reactivity /CO) for the 12 plumes including 11 western US wildfire plumes (circles) and 1 eastern US wildfire plume (square). Unweighted bivariate linear regression was applied to fit the data. The unweighted (or equally weighted) bivariate linear regression yields a slope = 0.31, $r^2 = 0.14 \pm 0.19$, and $p = 0.2$ for the 12 wildfire plumes. (b) Average secondary nHCHO production rate vs. total OH reactivity/CO for the 12 plumes including 11 western US wildfire plumes (circles) and 1 eastern US wildfire plume (square). An unweighted (or equally weighted) bivariate linear regression yields a slope = 0.32, $r^2 = 0.22 \pm 0.23$, and $p = 0.1$ for the 12 wildfire plumes.